# Impact of Fibrous Microplastic Pollution on Commercial Seafood and Consumer Health: A Review

**DOI:** 10.3390/ani13111736

**Published:** 2023-05-24

**Authors:** Serena Santonicola, Michela Volgare, Mariacristina Cocca, Giulia Dorigato, Valerio Giaccone, Giampaolo Colavita

**Affiliations:** 1Department of Medicine and Health Sciences “V. Tiberio”, University of Molise, 86100 Campobasso, Italy; colavita@unimol.it; 2Institute of Polymers, Composites and Biomaterials, National Research Council of Italy, Via Campi Flegrei 34, 80078 Pozzuoli, Italy; mariacristina.cocca@cnr.it; 3Department of Chemical Engineering, Materials, and Industrial Production, University of Naples Federico II, P. Tecchio 80, 80125 Naples, Italy; michela.volgare@unina.it; 4Independent Researcher, 37029 San Pietro in Cariano, Italy; dorigatogiulia@gmail.com; 5Department of Animal Medicine, Productions and Health, University of Padova, Viale dell’Università, 16, 35020 Legnaro, Italy; valerio.giaccone@unipd.it

**Keywords:** microplastics, microfiber pollution, seafood contamination, human exposure

## Abstract

**Simple Summary:**

Microfiber pollution is a widespread threat to marine fauna. These particles may be released into water from textiles during the washing process, and due to their low dimensions, the majority of microfibers cannot be blocked from wastewater treatment plants, reaching seas and oceans. Consequently, they could be ingested by marine organisms, including edible species, potentially leading to human exposure. However, microfiber and associated chemical exposure in fish and humans are still understudied. Further research is needed to better understand the potential negative impacts of microfibers on aquatic habitats, marine biota, and humans.

**Abstract:**

The omnipresence of microfibers in marine environments has raised concerns about their availability to aquatic biota, including commercial fish species. Due to their tiny size and wide distribution, microfibers may be ingested by wild-captured pelagic or benthic fish and farmed species. Humans are exposed via seafood consumption. Despite the fact that research on the impact of microfibers on marine biota is increasing, knowledge on their role in food security and safety is limited. The present review aims to examine the current knowledge about microfiber contamination in commercially relevant fish species, their impact on the marine food chain, and their probable threat to consumer health. The available information suggests that among the marine biota, edible species are also contaminated, but there is an urgent need to standardize data collection methods to assess the extent of microfiber occurrence in seafood. In this context, natural microfibers should also be investigated. A multidisciplinary approach to the microfiber issue that recognizes the interrelationship and connection of environmental health with that of animals and humans should be used, leading to the application of strategies to reduce microfiber pollution through the control of the sources and the development of remediation technologies.

## 1. Introduction

Microplastics have been defined as particles of different shapes (fragments, fibers, spheroids, granules, pellets, splinters, or beads) whose dimensions are between 0.1 μm and 5000 μm [1]. Furthermore, microplastics are classified into primary and secondary. Primary microplastics may result directly from the production of micro-sized particles for special domestic or industrial uses, while secondary microplastics may be derived from larger plastic objects by chemical or mechanical fragmentation [2]. Microplastics are highly persistent, accumulating in different marine habitats at increasing rates [3]. Among the different forms of microplastics found in the environment, several studies have shown that microparticles with fibrous shapes are predominant in the marine ecosystem [4,5,6], often accounting for more than 80% of the total items [7,8,9,10,11]. Microfibers are defined as “particles with a diameter less than 50 μm, length ranging from 1 μm to 5 mm, and length to diameter ratio greater than 100” [12]. Microfibers include both synthetic microfibers (e.g., nylon, polyester, polyolefin, and acrylic) and natural ones (e.g., cotton, flax, wool, and silk), which have been reported as the most abundant in the environment [6,12,13].

Microfiber pollution has become a global concern due to its wide diffusion not only in marine and freshwater habitats [14,15,16] but also in the air, soil, and sediments [16,17,18]. Most of the microfibers found in the oceans are released from textile industries [19]. Cotton is the most important natural fiber in the textile market and is second only to polyester, the predominant synthetic textile fiber [6]. Changes in fashion trends have, in fact, increased the demand for synthetic fabrics, mostly due to their low costs. In this context, Asia is the major synthetic fiber producer, and approximately 69% of total polyester fiber production is from China [12,19]. Domestic sewers and wastewater treatment plants (WWTPs) are considered the main pathways of textile microfibers in the marine environment [5,20]. A high number of microfibers may in fact be discharged from textile garments during domestic and industrial laundering processes [20,21]. In particular, more than 600,000 fibers may be released in a usual 5 kg wash load of polyester textiles [6,22]. Other sources that may contribute to microfiber pollution are fishing nets, curtains, carpets, and mattresses. On average, while 34.8% of microfibers in the oceans are derived from the laundering of synthetic textiles, 28.3% of microfibers are released from the friction of tires [6,12,19]. Moreover, due to the global COVID-19 epidemic, the wide consumption of masks, mainly composed of fiber materials that may be further broken to form microfibers, has also increased this type of pollution [23].

Once released into the environment, microfibers may be dangerous due to the risk of ingestion by marine species that are part of the food chain. Currently, microfiber exposure has been assessed both in farmed and wild fish species of commercial interest from all over the world [7,24,25,26,27,28,29]. As stated for microplastics, microfibers may also absorb, carry, and retain pollutants, which may be released in the tissues of animals ingesting these particles and lead to related health problems. Those chemicals might be more readily released from natural fibers because these may be degraded faster than synthetic ones [6]. However, the toxicological effects of natural microfibers remain understudied, and it is still unknown whether the impacts of these particles may differ from those of synthetic microfibers [30]. In addition, microfibers could act as important vectors of additives or dyes due to the textile processing, which may interact negatively with biota [11,19,31].

Considering the above, this review aimed to describe microfiber pollution in the marine environment, the impact on commercially relevant fish species and the marine food chain, and the probable threat to consumer health. For this purpose, a systematic literature search was performed, considering the occurrence of microfibers in commercially important fish species and the potential human health risks. The research was undertaken using academic search systems (PubMed, Web of Science, Scopus, and Google Scholar) and selecting a period from 2012 to February 2023 (Appendix A) in order to focus the attention on more recent available data.

## 2. Microfiber Pollution in the Marine Environment

Microfibers of natural or synthetic origin may enter the aquatic environment through different pathways, of which the wastewater discharged by textile industries and domestic laundering is the most important [12]. About 50% of microfibers from domestic drainage may escape the WWTPs and enter rivers and oceans. As a result, about 64,000 pounds of microfibers are released daily into the oceans [19]. Moreover, many microfibers may be formed directly in the sea due to the degradation of abandoned ropes and fishing nets [32].

Microfiber pollution was recognized in all major ocean basins [8,18,33] as well as within the marine trophic web [11,27,34,35,36,37,38]. In the Mediterranean Sea, microfibers released from synthetic fabrics represent about 40% (range 1.6–85.9%) of microplastics in seawater and the sea bottom, followed by fragments (mean 34.5%, range 1.6–72.7%) [11]. Most of the microfibers found in this area and in the Western Indian Ocean, North Atlantic, and South Atlantic Oceans were cellulose microfibers (79.5%) and animal microfibers (12.3%), such as wool [23]. In particular, microfibers are widely spread in estuaries and coastal waters [39], as documented in the Ebro Delta estuary in Spain [40] and in coastal waters in the Shanghai area [41], where synthetic microfibers represented 70% and 80%, respectively, of the total microplastics [42]. However, despite the fact that the majority of plastic pollution is derived from land and impacts coastal waters, recent evidence supports the hypothesis that currents may move these particles into the open ocean and to higher latitudes [8]. Considering the small size of microfibers, these may be more easily entrained in turbulent motions and flushed offshore to oceanic gyres towards the poles [43,44]. In particular, a mathematical model predicted that the accumulation of microfibers might respond to the role of surface water masses. These depend on the sea surface temperature and salinity, which show a pattern in the poles that could lead to microfiber accumulation [43]. Regarding the vertical distribution of microfibers in seawater, the physical shape and the nature of these particles may affect their position in the water column due to the different sinking rate densities and, therefore, their availability to marine organisms [23,45]. However, some aquatic species show dietary selectivity, which reduces the intake of microfibers. In contrast, larger aquatic organisms may ingest microfibers incorporated into their food, and omnivorous fish may be exposed to higher levels of microfibers than herbivorous and carnivorous species due to a wide range of food sources [23,46].

It has been observed that microplastic ingestion is associated with not only the size and shape of particles but also particle color. Blue, black, and transparent have been identified as the primary colors, especially for microfibers [44]. The extensive distribution of blue-colored microfibers in seawater has resulted in wide exposure in fish species due to active ingestion or indirect uptake, while transparent/clear (white and gray) microfibers may be ingested because they are mistaken for gelatinous prey [11,27].

Exposure to microfibers in marine organisms may cause physical damage, such as blockage in the gut, the segregation of digestive enzymes, low absorption of nutrients, disruption of the endocrine system, and disturbances in body functions, including respiration [6]. However, the effects of microfiber ingestion on marine biota could vary depending on the species and environmental conditions. As mentioned before, the chemicals absorbed by the microfibers and the leaching of toxic additives pose a significant threat to aquatic organisms [23,47]. Moreover, due to their large surface area and hydrophobic properties, synthetic microfibers can absorb hydrophobic pollutants and pathogenic microorganisms [48] that may enter the food chain [23].

Microplastic exposure in marine biota may also cause adverse consequences for biodiversity conservation, ecosystem services, and human food security (in terms of reduced food availability for the human population) [3,47,49]. Several health issues have been associated with microplastic ingestion in fish, such as the blocking of the digestive tract or a false sense of satiety. The evaluation of Fulton’s body condition, which is conducted by comparing the length and body mass, and the fullness index, which is conducted by quantifying the stomach content, may be useful for assessing a fish’s health status. Inflammation has also been reported, together with disturbances in the immune system and the metabolic profile [50,51]. Chronic exposure to microplastics may also be associated with behavioral changes and reductions in energy, growth, fecundity, and reproductive output [51,52]. Microfiber ingestion in oysters and mussels may negatively affect shellfishes’ reproductive output [53]. *Mytilus galloprovincialis* exposed to synthetic and natural microfibers resulted in different magnitudes of biological disturbance, depending on the nature of the microfiber [54]. Watts et al. [55] reported reductions in food consumption and in energy available for growth of the shore crab *Carcinus maenas* due to microfiber exposure. Slower feeding rates, as well as reduced reproductive output, have also been reported in planktonic copepod crustaceans, *Centropages typicus* [56], and *Calanus helgolandicus* [57], which are important food resources for a wide variety of fish [58]. Consequently, these individual and population impacts can cause ecosystem effects that would result in risks to food security in the near future [52]. However, there is a huge gap in this research field and a lack of information on the extent of this phenomenon. Few investigations have tried to examine some of these aspects, and discrepancies between laboratory and environmental conditions highlighted the need to consider the possible harmful effects related to microplastics via studies on wild-caught fish species [50].

## 3. Exposure to Microfibers in Fish Species for Human Consumption

### 3.1. Microfiber Contamination in Commercial Fish

Fisheries and aquaculture are crucial activities providing an important percentage of the worldwide food supply [59]. As global fish consumption continues to grow, understanding the potential threats of microplastic pollution is essential. Recently, more attention has been paid to the study of synthetic microfibers among the different types of microplastics, and several studies have revealed a considerable number of these particles in fish species of commercial interest, at levels higher than those reported for other microplastics (Table 1).

Due to their tiny size and wide distribution, microfibers may be ingested by wild-captured pelagic and benthic fish and farmed species [11,28,64]. The ingestion of microplastics by fish species may result from active foraging (i.e., mistaking plastic fragments and fibers for food) or from the incidental uptake from contaminated foods, sediments, or waters. Both mechanisms are important contributors to species-specific variations [25]. An investigation on microplastic exposure in different marine fish species (117 samples covering 39 species and 18 families) located in southern Taiwan confirmed that the major type of ingested microplastics was fiber (96%). Interestingly, no correlation was found between the trophic level and the number of ingested microplastics, suggesting that the potential biomagnification in fish is relatively small [7,66,67].

Different commercial fish species from the Adriatic Sea, which is considered a preferential area of plastic accumulation in the Mediterranean Sea, showed the occurrence of microfibers with frequencies of ingestion ranging between 40% and 70%. Most of the isolated microfibers were of natural origin (74% cotton, 8% wool) [64]. The ingestion of natural and synthetic microfibers has been assessed in European anchovy (*Engraulis encrasicolus*) and *Sardina pilchardus,* which are among the most captured fish species in the Mediterranean Sea. These species may be exposed to microfiber pollution because they are planktivorous and are mainly filter-feeding [29,68,69,70,71]. The microplastic contamination in anchovies from western Indonesia and South African waters confirmed that most of these particles had a fibrous shape, accounting, respectively, for 50.28 and 80% of ingested items [72,73].

A prevalence of fibrous microplastics was also observed in European hake (*Merluccius merluccius*) [24,25], an important commercial species in the Mediterranean Sea and the northeastern Atlantic Ocean. *M. merluccius* shows a mesopelagic behavior, representing an important link between pelagic and demersal habitats, and therefore has been proposed as a bioindicator to assess microplastic contamination in seawater [24]. The analyses of the gastrointestinal tracts of *M. merluccius* exemplars caught in the Adriatic Sea showed an average of 5.3 (±3.8) synthetic microfibers/gut [74], while the hakes collected from different FAO Geographical Sub-Areas of the Mediterranean Sea and from the Cantabrian Sea showed, respectively, contamination levels ranging from 0–8 and 8–26 particles/gut; among those, >80% were constituted by microfibers [24,75].

Similarly, red mullet (*Mullus barbatus*) a demersal fish species that lives in constant contact with sediment, has been designated as a sentinel species for several pollutants, including microplastics [24,76]. In this species, which swallows sediment together with prey, microfiber ingestion has been widely reported in samples collected from the Turkish shore, Adriatic and Tyrrhenian Seas, and the Mediterranean Spanish Coast [7,24,25,26,76]. The prevalence of microfibers (97.1%) among the ingested debris was also observed in other demersal fish species, such as the piper gurnard (*Trigla lyra*) and the blackmouth catshark (*Galeus melastomus*) from the Southern Tyrrhenian Sea. In particular, all the microfibers found in *T. lyra* were made of cellulose [11,26]. Microfibers were also the most abundant items in *Terapon jarbua*, *Ambassis dussumieri*, and *Mugil* spp. from the east coast of South Africa [77]. Interestingly, microbeads, pellets, and films were prevalent in three commercial fish species (*Sardinella maderensis, Dentex angolensis, Sardinella aurita*) from the Ghana Coast, where microfibers were the least occurring items (2%). Considering that domestic and industrial laundering processes have been identified as the main route of microfibers into the environment, the authors attributed these findings to the relatively low holding of washing machines in the West African sub-region [78].

Microfiber contamination was assessed in farmed fish (*Sparus aurata*) at different life stages at levels lower compared with wild fish [28]. Despite the strict control measures, the introduction of microfibers in aquaculture systems cannot be avoided. Analyses of the gastrointestinal tracts of 86 samples of farmed European sea bass (*Dicentrarchus labrax*, n = 45) and gilt-head sea bream (*S. aurata*, n = 41) from Tenerife (Canary Islands, Spain) showed the high prevalence of microfibers (100% and 96.1%, respectively), most of which were cellulose microfibers together with polyester, polyacrylonitrile, and poly(ether-urethane) [79]. Fishmeal has been investigated as a possible contamination source for farmed fish, showing an occurrence of about 124 microplastics per kg, including numerous microfibers (52.0 ± 14.0 items/kg), which may be derived from the contamination during the production process [80].

At the moment, the amount of research on the occurrence of microplastics, including microfibers, in edible muscle is very limited [81,82] compared to the number of studies investigating contamination in the gastrointestinal tracts of fish [3]. Microplastic contamination was observed in different brands of canned sardines and sprats [83], probably due to contamination during the canning process [84]. Zitouni et al. [85] reported the occurrence of microplastics smaller than 100 μm in the muscle tissues of a commercial fish species (*Serranus scriba*). These particles could be derived from the fragmentation of larger items in a fish’s digestive system and the translocation to other organs, including a fish’s edible parts [86,87]. On the other hand, microfibers could be particularly affected by this phenomenon because they are thin and may break into smaller pieces easily [88,89]. The differences in the sizes and shapes of microfibers could, therefore, influence translocation among fish tissues and human exposure through fish consumption [87].

### 3.2. Microfiber Contamination in Crustaceans

Shrimps, crabs, and lobsters are important crustaceans in commercial fisheries worldwide, considering their economic relevance [90]. To assess microplastic pollution, crustaceans have also been considered as possible bioindicators due to their filter-, suspension-, and deposit-feeding strategies [91,92,93]. However, at the moment, studies that assess microplastic uptake are still lacking for many commercial species [94,95,96], but the available information suggests that among the marine biota, crustaceans are also contaminated [97]. The study of microplastic abundance in the commercial shrimp *Pleoticus muelleri* showed that fibers were the predominant items (mean: 1.31 fibers/g wet weight) in the abdominal muscles [98]. Similarly, in other shrimp species, namely *Metapenaeus monocerus* and *Penaeus monodon*, synthetic microfibers were the most common microplastics in the gastrointestinal tract (57% and 32%, respectively) at levels ranging from 3.40–3.87 items/g [99]. Shrimps from the South China Sea showed a prevalence of polyester microfibers (70%) among the isolated debris [100], while nylon microfibers were observed in the stomachs of 5.93% *Plesionika narval* (narwhal shrimp) from the Aegean Sea [101]. Microfibers with varying size ranges and colors were predominantly also detected in decapod crustaceans [102,103]. Hara et al. [96] also found that microfibers were the main debris (98%) in *Nephrops norvegicus* from Irish waters.

The evaluation of microplastic contamination in 180 crabs (*Carcinus aestuarii*) from the northern Adriatic coast of Italy showed that only 5.5% of the samples contained microfibers, with a notable variability between individuals (from 1 to 117 particles per individual) that may be explained by the possible fragmentation of multifilament particles during the ingestion and passage through the gastrointestinal tract [104]. In addition, research on the congeneric species *C. maenas* showed that microplastics could be retained for long periods (>2 weeks), and the retention was even longer for microfibers [55]. As a result, crabs may play a critical role in the transfer of microfibers to higher trophic levels, including commercially relevant fish species that commonly prey on them, such as *Sparus aurata*, *Dicentrarchus labrax*, and *Anguilla anguilla* [104].

### 3.3. Microfiber Contamination in Bivalve Mollusks

Microplastic exposure in bivalves may depend on different factors, such as the species, biological parameters (e.g., size, filtration, and clearance rates), sampling period, and concentrations, types, and sizes of the plastic particles in the environment [105]. Mussels have been proposed as bioindicators, considering the positive correlation between the microplastic concentration in the surrounding water and bivalves, their wide distribution, low mobility, and commercial relevance [88,105,106,107]. Among the detected debris, different studies showed that fibers were the main shape identified in *Mytilus* spp. (Table 2).

Microplastic accumulation in mussels may be related to different mechanisms, such as ingestion, adherence, and fusion into the byssus [117,120]. Fibrous microplastics show a small diameter compatible with the feeding size range (around 15–30 μm) of filter-feeding organisms [121]. In addition, microfibers may be trapped in gills and hepatopancreases, resulting in a longer retention time compared with other microplastic types [110,122,123]. The high levels (from 1.43 to 7.64 microplastics/individual, 70.9% of which were fibers) of microfibers detected in mussels from the southwest of England were correlated to the longer retention time [123]. This fact should be taken into consideration when using mussels as bioindicators of microplastic pollution. Moreover, despite the fact that mussels are capable of eliminating some of the ingested microparticles, they may re-ingest them with consequent bioaccumulation in their tissues [122].

Among the different factors that may influence microplastic uptake, the mussel metabolic requirements, which show a typical seasonal pattern, should also be considered. There is a spawning period (spring and summer) followed by a gonad development phase (autumn and winter), during which weight loss is observed [123]. The currently available data showed no significant difference between the mussel sampling period and microfiber uptake [16,124], but deeper investigation is needed to better understand the physiological factors that determine the extent of microfiber exposure in bivalves. On the other hand, a negative correlation was observed between microfiber levels and mussel weight, which could be explained by the fact that in *Mytilus* species, pumping and filtration rates decrease with higher soft tissue mass [121,125].

Synthetic microfibers have also been reported as the dominant microplastic type in other commercial bivalve species [35,110,126,127,128,129], except for in the case of Cho et al. [130], where fragments were the most abundant. A study on microplastic content in the soft tissues of *Ruditapes decussatus*, *Cerastoderma* spp., and *Polititapes* spp. collected in Ria Formosa Lagoon in southern Portugal, showed a predominance (88%) of synthetic microfibers but no significant differences among species [131]. The assessment of microplastic contamination in wild populations of *Crassostrea gigas* from the Salish Sea, Washington State, revealed that 63% of oysters contained microparticles (~1.75 items/oyster), and microfibers were the dominant type of particles (96%) [132]. Similarly, the composition of microplastics in *Crassostrea virginica* samples from the Indian River Lagoon (Titusville, FL, USA) was dominantly microfibers (95%) [42]. Additionally, in *Aulacomya atra* samples sold in fisheries from three Peruvian provinces, the majority of the microplastics were of fibrous shape (58.8%) [133]. Usually, farmed bivalves contain more microfibers than wild specimens since they grow on polypropylene lines and are often cultured in coastal areas [126,129]. The depuration that they can undergo before being sold may help them to eliminate filtered microplastic, and several studies investigated this phenomenon using depuration experiments [122,134,135,136]. The results showed that about 30–40% of synthetic microfibers were eliminated after 48 h of depuration, but small microfibers (50 µm–1 mm) were less affected by depuration and were more accumulated than larger ones (1–5 mm) [120]. Interestingly, synthetic blue microfibers were found to be the type of microplastics that were more effectively eliminated during the depuration process (96 h), with a decrease of 46.79% for wild mussels and 28.95% for farmed mussels, which seems to imply a preferential elimination of particles with a specific color, probably due to the additives or pigments employed [122].

## 4. Detection Methods of Microfibers in Bivalve Mollusks and Fishes

To assess the microfiber uptake of marine organisms, the common procedures used for the detection of microplastics in marine biota may be employed [35]. However, microfibers were frequently excluded from the results because of methodological issues understating the effective contamination. Therefore, over the years, stricter precautions have been applied during sample processing in order to reduce secondary and airborne contamination [11,137,138,139]. Laboratory blanks that correct this contamination are especially important for microfibers [30].

The gastrointestinal tracts extracted from fish species and bivalve soft tissues are usually digested using different chemical solutions (i.e., hydrogen peroxide, potassium hydroxide, nitric acid) [94] or by enzymes (i.e., cellulase, proteinase-K, or chitinase) [140,141]. Considering that long digestion time (>4 days) using potassium hydroxide may cause microfiber morphological alterations, enzymatic extraction has been proposed as an alternative in microfiber analyses for larger organisms requiring longer digestion times [104]. A comprehensive evaluation of microfiber exposure in seafood should also include analyses of natural microfibers [64]. However, not all methods are suitable to isolate these particles since certain chemical digestants may cause the degradation of non-synthetic microfibers [30].

The small size of these particles and the presence of dyes on the microfibers may hamper the identification using conventional spectroscopic techniques [142,143]. Moreover, the choice of appropriate substrates represents an important drawback, considering that cellulose membranes, which are frequently employed for the filtration of biological matrices, may interfere with signal detection and spectra acquisition [121]. Another important challenge is the low spectral signal intensities of natural materials, which make them more susceptible to dye interference [30]. In this context, the morphological analysis of the microfibers based on the evaluation of surface morphology (i.e., shape and texture) has been proposed as an alternative approach in order to identify natural and synthetic microfibers and ascertain their origin [76,121,142]. Natural microfibers usually appear twisted like flat ribbons and do not show a uniform diameter, while synthetic fibers show a cylindrical cross-section and smooth and shiny surface [29,121]. In more detail, cotton may be recognized by its flattened and curly shape [76,144], while synthetic microfibers are long and smooth [140]. The occurrence of numerous natural microfibers in commercial fish species highlights the importance of successfully distinguishing between synthetic and natural microfibers, considering that the latter are rarely documented and not included when assessing the impact of microfiber pollution on marine biota [37].

## 5. Potential Consumer Health Risk

According to today’s knowledge on microfibers in seafood, mainly bivalves, which are usually eaten whole, could contribute to the amount of ingested microplastics by humans [125]. However, small pelagic fish species that could be consumed whole (e.g., sardines, anchovies, and sprats) may also cause consumer exposure [71,145] (Figure 1).

In other commercial fish species and crustaceans, the presence of microfibers has, in most cases, been investigated only in the gastrointestinal tract [64,99], but the translocation of these particles in edible tissues may occur [86,87,89]. Some studies have in fact shown the occurrence of microfibers in the gutted meat of some marine organisms at levels even higher than those in viscera [3]. Moreover, fish from point-of-sale may undergo additional contamination due to airborne fallout from clothing and machinery during processing or from packaging [29,146]. Therefore, those who consume processed seafood may not be able to avoid microfiber pollution effectively [23].

Moreover, in different regions, such as East and West Africa, where the consumption of fish, including the gastrointestinal tract, is more common, the accidental ingestion of plastics has raised major concerns [78]. In this respect, the microplastic intake by humans depends not only on the microplastic content in seafood but also on eating habits that may vary significantly among regions [49]. The average Korean may be exposed to 212 particles/person annually via the consumption of oysters, mussels, manila clams, and scallops [130]. European minor shellfish consumers may ingest around 1800 microplastics annually, while top shellfish consumers may be exposed to up to 11,000 microplastics per year [127]. In general, Cox et al. [147] evaluated that about 39,000–52,000 microplastics could be ingested via food consumption (e.g., seafood, sugar, and honey) per person per year. Moreover, consumers may be exposed to as many as 5800 synthetic fibers per year from tap water, beer, and sea salt; among those, tap water contributes 88% [148].

However, according to Catarino et al. [140], microplastic ingestion through contaminated food is minimal compared to airborne household fibers that fall into our meals. In fact, a large number of fibrous microplastics (13,731–68,415 per year) are ingested during meals via household dust fibers, over 10 times more than that via mussel consumption (123–4620 particles/year) [125].

Once ingested, microfibers may penetrate the human body via cellular uptake or paracellular transport in the gut, and the degree of uptake may vary according to the shape, size, and chemical composition. Particles on the scale of a few microns and up to 10 μm may be, respectively, taken up by cells in the lungs or gut and by specialized cells in the Peyer’s patch of the ileum, while those as large as 130 μm can enter tissue through paracellular transport [147].

The main concerns with microplastics, including microfibers, are that they may carry potentially harmful chemicals, such as plastic monomers and additives, or adsorb toxic pollutants from the marine environment [84]. The fibrous particles and the associated chemicals may be transferred through the food chain with detrimental consequences to humans. Phthalates, which are used in some steps of the textile production process or may interact with microfiber as water pollutants, have been proven to act as endocrine disruptors and lead to breast cancer and damage of the liver, kidneys, and intestines [149,150]. Additionally, Bisphenol A (BPA), another chemical linked to the textile industry, may impact human health by disrupting endocrine functions [19]. However, based on the current knowledge, the contributions of these chemicals from microplastics among top consumers are very small compared to other sources [84]. Microfibers may also act as a vehicle for pathogenic microorganisms that can form biofilms on their surface when they come into contact with wastewater. These pathogens may be released by microfibers in bathing or drinking water, resulting in human exposure, but little information is available on this threat, and more knowledge on the microbial community composition of microplastic biofilms is required [6,151].

Despite the fact that exposure to microplastics, including synthetic microfibers, may pose significant concerns, the required quality-controlled data to make a safety risk assessment are lacking [84]. A comprehensive exposure assessment is not applicable, considering the lack of representative data on the occurrence of microplastics in different food groups [152]. Moreover, in the absence of a reference value of a tolerable microplastic intake for humans through the ingestion of seafood, it is not possible to perform a risk assessment of human exposure [153]. As discussed before, microfibers are underestimated in the literature. Given the uncertainty and potential misidentification of fibrous particles, further work is needed to fully assess natural and synthetic microfiber pollution [30]. Future studies based on consistent sampling and analysis methods of seafood from markets [154] could help to implement monitoring programs to assess microfiber contamination. More information is needed on microfiber pollution in marine species captured for consumption, with a special focus on those eaten whole as well as seafood sold in fishery markets and ready for consumption, considering that the microplastic and microfiber contamination in shellfish and seafood may be also associated with the handling and supply chain [133].

## 6. Knowledge Gaps and Future Challenges

Despite the fact that microfibers have become increasingly ubiquitous all over the world, microfiber research is still in its infancy, and there are many questions to be answered. The potentially negative impacts of microfibers on aquatic habitats, marine organisms, and humans are still unknown. The long-term exposure of microfibers and their associated chemicals on fish and humans are still understudied, but they should be considered a research priority [12], as plastic-derived chemicals have also been detected in fish muscle [155]. There is an urgent need to standardize data collection methods for the detection of microfibers in the environment and foodstuffs, followed by exposure assessments for dietary intake [156]. In this context, natural microfibers should also be investigated rather than overlooked because of their biodegradability [6], considering that the degradation process could be considerably delayed due to the use additives and dyes, such as flame retardants [23].

Further focus on freshwater fish and aquaculture seafood to understand the potential sources of microfiber presence in these environments should be given [154]. In addition, future research needs to include other food groups, such as grains, vegetables, beef, and poultry, which represent major sources of nutrition globally [147], and identify processing and cooking methods to promote adjustments rather than consumer avoidance of certain food categories [156]. For example, it has been estimated that humans’ microfiber intake can be reduced by about 14% in the case of cooked mussels if the cooking water is not consumed [110].

The issue of microfiber pollution can be addressed at multiple levels, involving clothing companies, factories, consumers, and municipal wastewater treatment plants. More research into fabrics that are more environmentally friendly and innovative solutions are needed in order to reduce the microfiber entering the oceans [19]. In this context, a “One Health approach” to the microfiber issue that recognizes the interrelationship and connection of environmental health with that of animals and humans is necessary, considering that microfiber pollution may impact aquatic ecosystems and biota, resulting in risks also to food security [52,157]. This approach may lead to the application of strategies to reduce microfiber pollution focused on effectively controlling the sources and the development of remediation technologies [158,159].

## 7. Conclusions

The occurrence of microfibers in commercially relevant seafood with different habitats and feeding strategies confirms their wide distribution in the aquatic environment. The available findings point to the need to assess the risks of microfibers on marine food chain and their potential threat to human health as priority research areas. Limited knowledge on the role that microfibers is playing on food security and safety is available, and more attention should be given to understanding the mechanism of microfiber translocation in marine biota from gut to internal tissues, which may result in the transfer via consumption to high trophic levels and humans.

## Figures and Tables

**Figure 1 animals-13-01736-f001:**
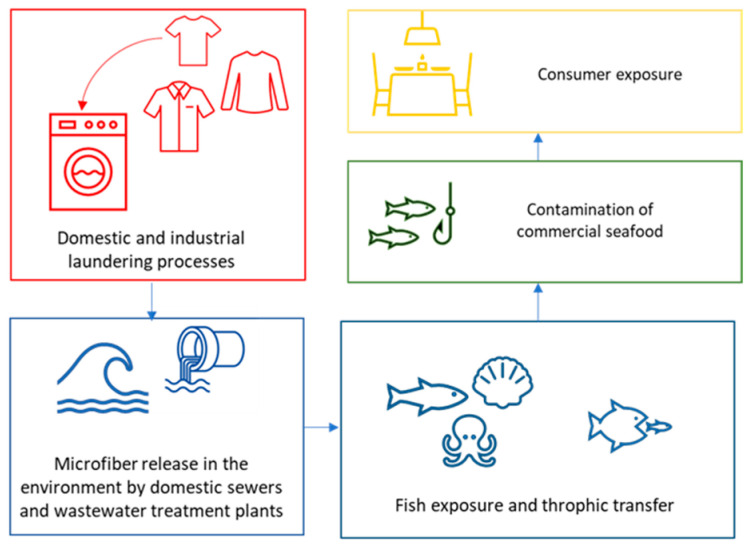
Microfiber transfer through the marine food web and human exposure.

**Table 1 animals-13-01736-t001:** Synthetic microfiber (MF) percentage among the isolated microplastics (MPs) in commercial fish species.

Species	Location	MPs/Gastrointestinal Tract *	% MFs	Reference
*Merluccius merluccius* *Mullus barbatus*	Spain	1.56 ± 0.5	71%	[25]
*Acanthopagrus australis* *Mugil cephalus* *Gerres subfasciatus*	Sydney Harbour, Australia	1.6 ± 0.8 4.6 ± 1.2 0.2 ± 0.1	83%	[3]
*Mugil cephalus*	South Africa	3.8 ± 4.7	51.2%	[60]
*Galeus melastomus*	Western Mediterranean Sea	0.34 ± 0.07	86.4%	[61]
*Ammodytes personatus* *Clupea pallasii*	USA	1–9 ** 5–27 **	100%	[62]
*Pagellus erythrinus* *P. bogaraveo*	Tyrrhenian Sea	-	100%	[63]
*Sardina pilchardus* *Scomber scombrus* *Trachurus trachurus* *Solea solea*	Northern Adriatic Sea	1.4 ± 0.55 ^a^–3.67 ± 2.06 ^b^ 1.3 ± 0.58 ^a^–4.22 ± 1.71 ^b^ 2 ± 0 ^b^ 1 ± 0 ^a^–2 ± 1.41 ^b^	-	[64]
*Engraulis encrasicolus*	Ligurian Sea	0.12 ± 0.12 ^a^–0.34 ± 0.29 ^b^	-	[27]
*Trachurus murphyi* *Strangomera bentincki* *Merluccius gayi*	Chile	- ***	70–100%	[65]
*Sparus aurata* *Cyprinus carpio*	Fish farms located in Italy and Croatia	0.48 0.11	~90%	[28]
*Mullus barbatus* *Trigla lyra* *Galeus melastomus* *Scyliorhinus canicula* *Raya miraletus*	Southern coasts of Sicily	0.3 0.4 0.1 1.1 2.0	97.1%	[26]

* mean value, ** range, *** only the total amount (n = 20) of detected microplastics was reported, ^a^ microplastics/gastrointestinal tract, ^b^ microfibers/gastrointestinal tract.

**Table 2 animals-13-01736-t002:** Percentage of synthetic microfibers (MFs) among the isolated microplastics (MPs) in *Mytilus* spp.

Species	Location	MPs g/ww *	MPs/Individual **	% MFs	References
*M. galloprovincialis*	Italy	-	1–2	61–100%	[108]
*M. galloprovincialis*	China	2	0.53	84.11%	[109]
*M. galloprovincialis*	Italy	4.4–11.4	3.0–12.4	100%	[110]
*M. galloprovincialis*	Tunisia	0.8	-	91.3%	[111]
*M. galloprovincialis*	Italy, Netherlands	-	0.03 ± 0.03	98.5%	[112]
*M. galloprovincialis*	South Africa	2.8	3.4	67%	[113]
*M. edulis*	Belgium	0.21–0.51	-	100%	[114]
*M. edulis*	Netherlands	-	5–19	24%	[95]
*M. edulis*	China	1.52–5.36	0.77–8.22	86%	[107]
*M. edulis*	United Kingdom	-	1.43–7.64	87%	[115]
*Mytilus* spp.	Portugal, Italy, Spain, France, Denmark	0.13–0.18	-	82.6%	[116]
*Mytilus* spp.	China	-	0.85–1.2	55–68%	[117]
*Mytilus* spp.	Spain	0–8.9	0–10	63–68%	[118]
*Mytilus* spp.	USA	-	0.9 ± 0.6	96%	[119]

* ww: wet weight, ** mean value.

## Data Availability

No new data were created or analyzed in this study. Data sharing is not applicable to this article.

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
