# Peer review of "Impact of Fibrous Microplastic Pollution on Commercial Seafood and Consumer Health: A Review"

_animals, 2023, doi:10.3390/ani13111736_

Round 1
Reviewer 1 Report
The article "Impact of fibrous microplastic pollution on commercial fish 2 species and consumer health. A review" well summarizes the most recent publications in this field. The manuscript is well-written and enjoyable to read. Although there are countless reviews on the subject of microplastics, I believe this manuscript is suitable for publication.
I only have very few and minor suggestions:
Line 18: I would suggest changing it to “long-term exposure to microfiber and associated ….”
Line 54: is it really assessed that most of the microfibers in the oceans come from textile industries? If it is so, please add an appropriate reference. Or it was meant that they come from textiles during their post-production life?
Lines 129-143: research on the effects of microplastic exposition on aquatic organisms has reached many more results than what is presented here. I would suggest referring more generally also to some reviews (for example 10.26650/ASE20221186783, 10.1016/j.ecoenv.2019.10991, 10.3390/w13162214)
Line 324: Since natural fibres are not the focus of this review I would suggest rephrasing this sentence
Author Response
The point-by-point responses to reviewer 1 comment are in the attached file.

Reviewer 2 Report
General Comment
The submitted ms reviews the microplastic(MP) pollution of commercial fish species on consumer health. The text is well structured and easy to read. I proposed only some alterations in the specific comments list below. A major concern is related to the relation of fiber contamination and human health. Most of the references cited analysed MP and fibers in the gastrointestinal tracts of fish. Since usually muscle is consumed, the authors should state clearly what tissue was analysed. Results based on counts of MP the intestines, should not be reported as number of MP /individual. Most probably individuals with MP in the guts will show MP as well in muscle, liver, kidney etc. The same applies to other organisms like molluscs or crustaceans. The unit of reporting contamination is important. Example: 10 fibers/g muscle.
Line Specific comments
1 The title should include seafood in general, since the authors refer to fish, crustaceans and molluscs.
21 Omnipresence may be more appropriate than “prevalence”
23 “…by wild captured pelagic or benthic fish and farmed species.”
24 Humans are exposed by seafood consumption.
25 “…knowledge on our role…. is limited.”
29 “…edible species are under threat….” I is not clear what threat you mean.
71 Although the contamination of the food chain is common language use, it is not technically correct. Contamination is measured by number of fibers / volume or weight etc. The food chain cannot be described by these units. However, parts or members of the food chain can…
75-76 These interesting aspects of natural fibers. They are not frequently discussed.
88-89 Both mentioned pathways are very similar. Suggestion: “… may enter the aquatic environment through different pathways of which wastewater discharged by….. is the most important.
93 …abandoned ropes and fishing nets.
99 What are animal microfibers? Can you be more specific?
107 “…flushed to the oceanic gyres” o.k. but why to the poles?
117-119 Please rephase the difference between transparent and clear fibers. Blue fibers are only exposed, while other are ingested?
153 Table 1 refers to microfiber percentage among isolated microplastics in commercial fish species. As the review deals with microplastic pollution and human health effects the authors should include the information, in which tissue the particles were found. When checking the cited references, it appears to me that all are related to the presence of microplastics in the gastrointestinal tracts. The appropriate unit to show these results is therefore number of particles or fibers/tract and not per individual. The number of particles per individual most probably is much higher, since other tissues where not analysed. Possible human health effects are most probably related to the consumption of muscle.
158-159 …may be ingested by wild captured pelagic or benthic fish and farmed species.
182-188 Again the problem of units: Particles/individual is not correct. The authors should state clearly which tissues were analysed.
191 ….that swallows sediment…..
198, 207 I find it confusing if the authors mix farmed freshwater species like tilapia or carps with marine species. Since the contamination of wild freshwater species is not the subject of the paper, farmed freshwater species should not be included either.
235 …crustaceans are also contaminated.
376 …during dinner? Is that correct?
The text is well structured and easy to understand.
Author Response
The point-by-point responses to reviewer 2 comments are in the attached file.

Round 2
Reviewer 2 Report
The revised version addressed my questions and proposals thoroughly.
Only one additional observation: Table 2, column 4 line 3: Most probably the value of 43 should be 4.3.
An additional round of reviewing is not necessary.
No further comments.
Good luck for the authors
Author Response
The authors thank the reviewer for the valuable suggestions.
The values in Table 2 (column 4, line 3) were corrected.